# Ethanol Extract of Radix Asteris Suppresses Osteoclast Differentiation and Alleviates Osteoporosis

**DOI:** 10.3390/ijms242216526

**Published:** 2023-11-20

**Authors:** Sung-Ju Lee, Hyun Yang, Seong Cheol Kim, Dong Ryun Gu, Jin Ah Ryuk, Seon-A Jang, Hyunil Ha

**Affiliations:** 1KM Convergence Research Division, Korea Institute of Oriental Medicine, Yuseong-daero 1672, Yuseong-gu, Daejeon 34054, Republic of Korea; sungjulee@kiom.re.kr (S.-J.L.); hyunyang@kiom.re.kr (H.Y.); iron0907@kiom.re.kr (S.C.K.); mrwonsin@kiom.re.kr (D.R.G.); yukjinah@kiom.re.kr (J.A.R.); 2Future Technology Research Center, KT&G Corporation, 30, Gajeong-ro, Yuseong-gu, Daejeon 34128, Republic of Korea; jsa85@ktng.com

**Keywords:** Radix Asteris, osteoporosis, ovariectomy, osteoclast

## Abstract

Radix Asteris, the root of *Aster tataricus* L. f., is historically significant in East Asian medicine for treating respiratory conditions. Yet, its implications on bone health remain uncharted. This research investigated the impact of an aqueous ethanol extract of Radix Asteris (EERA) on osteoclast differentiation and its prospective contribution to osteoporosis management. We discerned that EERA retards osteoclast differentiation by inhibiting receptor activator of nuclear factor kappa-B ligand (RANKL) expression and obstructing RANKL-induced osteoclastogenesis. EERA markedly suppressed RANKL-induced expression of NFATc1, a pivotal osteoclastogenic factor, via modulating early RANK signaling. EERA’s therapeutic potential was underscored by its defense against trabecular bone degradation and its counteraction to increased body and perigonadal fat in ovariectomized mice, mirroring postmenopausal physiological changes. In the phytochemical analysis of EERA, we identified several constituents recognized for their roles in regulating bone and fat metabolism. Collectively, our findings emphasize the potential of EERA in osteoclast differentiation modulation and in the management of osteoporosis and associated metabolic changes following estrogen depletion, suggesting its suitability as an alternative therapeutic strategy for postmenopausal osteoporosis intertwined with metabolic imbalances.

## 1. Introduction

Bone remodeling is a crucial physiological mechanism essential for lifelong skeletal maintenance and repair. This process is steered by the synergistic actions of osteoblasts, which facilitate bone formation, and osteoclasts, tasked with bone resorption [1,2]. Positioned within the bone matrix, osteocytes serve as the key regulators of bone remodeling, adept at detecting mechanical stresses and liaising with osteoblasts and osteoclasts to oversee their functions. Disruptions in the harmonious interplay of these cells can precipitate various skeletal disorders. Notably, postmenopausal osteoporosis emerges as a predominant disorder, hallmarked by diminished bone density and an increased susceptibility to fractures. This condition largely stems from estrogen decline following menopause, which amplifies osteoclast activity, resulting in accelerated bone degradation [1,3].

Osteoclasts are specialized multinucleated cells that arise from the monocyte/macrophage lineage and are primarily responsible for bone resorption [2]. Their differentiation is governed by a series of complex molecular pathways. A central regulator in this process is RANKL [2]. Although RANKL expression is observed in diverse cell types, it is predominantly produced by osteocytes within the bone matrix during remodeling [4,5]. RANKL binding its receptor osteoclast precursors initiates a series of intracellular signaling cascades by recruitment of the adaptor molecule tumor necrosis factor receptor-associated factor 6, steering these precursors toward their maturation into bone-resorbing osteoclasts [6,7]. The transcription factor, nuclear factor of activated T cells cytoplasmic 1 (NFATc1), is integral to this RANKL-induced differentiation [8]. Its expression and activity are modulated by various molecules, including c-Fos, a key transcription factor that operates upstream of NFATc1 [9]. Conversely, the induction of B lymphocyte-induced maturation protein-1 (Blimp1) by NFATc1 enhances NFATc1 expression through the transcriptional repression of antiosteoclastic genes [10].

Given the central importance of the RANKL/RANK pathway in osteoclast differentiation and its implications in pathological bone loss, interventions aimed at modulating RANKL expression and RANK signaling have been identified as promising strategies to counteract excessive bone deterioration [3]. While contemporary therapeutic approaches, including antiresorptive drugs targeting RANKL and osteoclast function, are available, concerns regarding their prolonged use and potential adverse effects have amplified interest in exploring natural remedies. This encompasses dietary foods, medicinal plants, and their active constituents, which have shown beneficial impacts on the differentiation and function of osteoclasts and osteoblasts [11,12].

*Aster tataricus* L. f., a member of the Asteraceae family, is a perennial herb native to regions such as northeast and northwest China, Korea, Japan, and eastern Siberia. The dried root of this plant, known as Radix Asteris or “Ziwan”, has been historically integrated into medicinal practices across East Asia, particularly within traditional Chinese medicine and Korean medicine. Its therapeutic applications are believed to include lung hydration, alleviation of cough, and promotion of phlegm clearance [13]. Recent pharmacological studies have corroborated some of these traditional claims, highlighting the extract’s expectorant and antitussive properties [14], its role in attenuating asthma-related symptoms [15], and its capacity to counteract acute lung injuries [16]. Furthermore, benefits extend to conditions like constipation [17], benign prostatic hyperplasia [18], and diabetic retinal damage [19], attributed largely to its anti-inflammatory and antioxidant properties. Despite these findings, the potential effects of Radix Asteris extract on bone health have not been researched. This study aimed to elucidate the potential of EERA on osteoclast differentiation in vitro and evaluate its prospective therapeutic implications for osteoporosis and other metabolic changes in ovariectomized (OVX) mice.

## 2. Results

### 2.1. EERA Suppresses Osteoclast Differentiation in an Osteoclast Precursor–Osteocyte Coculture System

To evaluate the influence of EERA on osteoclast differentiation, a coculture system was established using osteoclast precursors (bone-marrow-derived macrophages, BMMs) and osteocytic MLO-Y4 cells. Previous studies have demonstrated that in coculture conditions, MLO-Y4 cells act as osteoclast-supporting cells by presenting surface-bound RANKL [20]. 1α,25-dihydroxyvitamin D3 (VitD3) has been shown to augment the ability of these cells to promote osteoclastogenesis by upregulating RANKL and concurrently downregulating its decoy receptor, osteoprotegerin (OPG) [20,21]. Post a 5-day treatment with 10 nM VitD3, osteoclast differentiation was evident, characterized by the emergence of large tartrate-resistant acid phosphatase (TRAP)-positive multinucleated cells. Notably, a dose-dependent inhibition of osteoclastogenesis was observed with EERA, with complete suppression manifested at 200 μg/mL (Figure 1A, upper panel).

Further analysis of MLO-Y4 cells revealed alterations in gene expression. Treatment with VitD3 resulted in elevated *Tnfsf11* (encoding RANKL) and suppressed *Tnfrsf11b* (encoding OPG) mRNA levels, while Csf1 (encoding macrophage-colony-stimulating factor, M-CSF) gene expression remained stable. EERA significantly curtailed the VitD3-induced RANKL upregulation without impacting OPG and M-CSF levels (Figure 1B). However, EERA alone did induce a decline in OPG expression. Consistent with the observed decrease in RANKL mRNA levels, EERA markedly diminished both the basal and VitD3-stimulated RANKL protein expression in MLO-Y4 cell lysates (Figure 1C). These findings suggest that the inhibitory effect of EERA on osteoclast differentiation is attributed to its ability to suppress RANKL expression in MLO-Y4 cells. To discern whether the RANKL reduction was the sole factor behind EERA’s inhibitory effect, exogenous RANKL and VitD3 were introduced to the coculture system with and without EERA (Figure 1A, lower panel). The presence of exogenous RANKL failed to counteract EERA’s suppressive effect on osteoclastogenesis, hinting at additional inhibitory pathways at play.

### 2.2. EERA Inhibits RANKL-Driven Osteoclastogenesis

To delineate EERA’s direct influence on osteoclast differentiation without the involvement of MLO-Y4’s osteoclast-supportive role, we exposed BMMs to RANKL for 4 days, excluding MLO-Y4 cells from the culture. Echoing the outcomes from our coculture system, EERA consistently demonstrated a dose-responsive attenuation of RANKL-stimulated osteoclast differentiation (Figure 2A,B). Importantly, EERA did not exhibit cytotoxic effects on BMMs; on the contrary, it enhanced cell viability (Figure 2C), indicating that the antiosteoclastogenic effect of EERA is not due to cytotoxicity on osteoclast precursor cells. Cumulatively, these findings suggest that EERA directly impedes osteoclast differentiation by targeting osteoclast precursors, in addition to its indirect role in downregulating RANKL expression.

### 2.3. EERA Modulates RANKL-Driven Signaling Pathway

To understand the mechanistic basis of EERA’s inhibitory effects on RANKL-mediated osteoclastogenesis, we analyzed key regulatory factors. It is known that RANKL promotes osteoclastogenesis by enhancing the expression of the pivotal osteoclastogenic transcription factor, NFATc1 [8]. Following BMM treatment with EERA, there was a discernible attenuation in the RANKL-stimulated mRNA and protein levels of NFATc1 during osteoclastogenesis (Figure 3A,B). Additionally, EERA curtailed the expression of NFATc1 downstream osteoclastogenic genes, including *ATP6v0d2* and *Tm7sf4* [22,23,24], crucial for osteoclast fusion, as well as *Ctsk* [24,25], implicated in osteoclast-driven organic bone matrix degradation (Figure 3B).

Notably, c-Fos functions as an upstream transcription factor for NFATc1 during osteoclastogenesis [9]. EERA treatment resulted in diminished c-Fos protein levels, yet its mRNA expression remained unaffected during osteoclastogenesis (Figure 3A,B). Further, it has been established that RANKL elevates NFATc1 expression by suppressing its transcriptional repressor genes, such as *Irf8* [26]. Blimp1, encoded by *Prdm1* gene, further modulates these repressors [10]. In our study, EERA was found to mitigate the RANKL-stimulated elevation of *Prdm1* while preserving *Irf8* levels (Figure 3B).

RANKL activates critical early signaling cascades, encompassing MAPKs and the NF-κB pathway, central to NFATc1 expression and osteoclast differentiation [27,28,29,30]. EERA treatment inhibited RANKL-driven activation of JNK and p38 MAPKs but amplified ERK activation. RANKL treatment resulted in the phosphorylation followed by degradation of IκBα, integral to classical NF-κB pathway activation. Notably, while EERA attenuated the degradation of IκBα, its phosphorylation was persistent (Figure 3C).

Taken together, our data suggest that EERA suppresses NFATc1 expression and osteoclastogenesis by modulating both the positive regulator c-Fos and the inhibitory regulator Irf8, likely mediated via alterations in RANKL-induced early signaling pathways.

### 2.4. EERA Administration Attenuates Bone Loss and Fat Accumulation in OVX Mice

We postulated the therapeutic benefits of EERA from its in vitro antiosteoclastic properties and validated it using OVX mice—a well-established model for postmenopausal osteoporosis. Postovariectomy, mice received EERA orally at either 30 mg/kg/day (EERA-30) or 100 mg/kg/day (EERA-100) over 6 weeks. Microcomputed tomography (μ-CT) analysis of trabecular bone in the distal femur metaphysis revealed pronounced trabecular bone loss in OVX mice compared with sham controls. Nevertheless, both EERA-treated groups showed significant mitigation of this loss (Figure 4A). Quantitatively, compared with sham controls, the OVX group showed a 24.8% decrease in bone mineral density (BMD), a 49.8% reduction in bone volume per tissue volume (BV/TV), and a 50.8% decline in trabecular number (Tb.N), along with a 91% increase in trabecular separation (Tb.Sp). EERA treatment reversed these deteriorations, with both dosages showing comparable osteoprotective efficacy (Figure 4B). Notably, EERA reduced the ovariectomy-induced BMD decline by up to 44.6%, BV/TV dropped by up to 49.5%, Tb.N decreased by up to 37.3%, and Tb.Sp increased by up to 40.9%.

Beyond bone metrics, OVX mice typically undergo physiological shifts: increased body weight, augmented perigonadal fat, liver hepatic steatosis, and uterine atrophy [31]. Additionally, estrogen deficiency also leads to the enlargement of immune organs, especially the spleen and thymus [32]. Our study revealed EERA’s potential to counter ovariectomy-induced weight and fat gain; however, fat reduction at 30 mg/kg/day was statistically analogous to OVX controls (Figure 5A). Thymus weight increment and spleen weight trends observed in OVX mice were significantly curbed with EERA (Figure 5A). Moreover, while OVX mice exhibited raised serum alanine transaminase (ALT) levels hinting at liver damage, EERA at 100 mg/kg/day significantly ameliorated this. Serum aspartate transaminase (AST) patterns paralleled ALT, but no significant differences were discerned across the groups (Figure 5B). Notably, EERA administration did not alter the uterine atrophy evident in OVX mice (Figure 5A). This indicates that EERA’s beneficial effects on bone and metabolic alterations in OVX mice are unlikely attributed to estrogenic actions.

### 2.5. Phytochemical Characterization of EERA

Through ultrahigh-performance liquid chromatography–tandem mass spectrometry (UHPLC-MS/MS), we characterized the phytochemical constituents of EERA. Previous investigations have identified a diverse array of phytochemicals in Radix Asteris, predominantly encompassing terpenes, peptides, organic acids such as chlorogenic acids, and flavonoids [13]. Consistently, our analysis of EERA detected four chlorogenic acids (chlorogenic acid, feruloylquinic acids 1 and 2, and 3,4-dicaffeoylquinic acid), a triterpenoid saponin (astersaponin A), six peptides (astin A, C, E, and J, asterinin A, and iso-asterinin A), two flavonoids (kaempferol and quercetin), and three long-chain fatty acids (pinellic acid, linoleic acid, and oleic acid). This profile is illustrated in Figure 6 and detailed in Table 1.

It is noteworthy that several phytochemicals within EERA have established roles in bone and metabolic health. Chlorogenic acid, for instance, has demonstrated its efficacy against ovariectomy-induced osteoporosis, promoting osteoblast differentiation and resisting osteoclast differentiation [33,34]. Furthermore, it inhibits adipocyte differentiation and curtails weight gain and fat accumulation in mice on a high-fat diet [35,36]. 3,4-dicaffeoylquinic acid exhibits an inhibitory effect on both osteoclast and adipocyte differentiation [37,38]. Quercetin displays protective attributes against ovariectomy-induced osteoporosis in rats, favoring osteogenic differentiation while curtailing RANKL expression [39,40]. Additionally, kaempferol alleviates ovariectomy-induced osteoporosis, likely by suppressing osteoclastic bone resorption and enhancing osteoblastic differentiation while also reducing ovariectomy-induced weight gain [41,42].

**Table 1 ijms-24-16526-t001:** UHPLC-MS/MS analysis of EERA phytochemicals.

Peak No.	R.T. (Min)	Ion Mode	Error (ppm)	Formula	Expected Mass (*m/z*)	Measured Mass (*m/z*)	MS/MS Fragments (*m/z*)	Identification	Reference
1	5.15	${\left[M-H\right]}^{-}$	0.142	C_16_H_18_O_9_	353.0878	353.0879	191.0553, 179.0338, 161.0234, 135.0437	Chlorogenic acid *	[43]
2	5.43	${\left[M-H\right]}^{-}$	2.564	C_17_H_20_O_9_	367.1012	367.1033	193.0498, 173.0445, 134.0360	One of feruloylquinic acid	[14]
3	6.18	${\left[M-H\right]}^{-}$	3.109	C_17_H_20_O_9_	367.1012	367.1035	193.0497, 173.0444, 137.0230	One of feruloylquinic acid	[14]
4	7.39	${\left[M-H\right]}^{-}$	0.342	C_25_H_24_O_12_	515.1195	515.1193	191.0554, 179.0340, 161.0233, 135.0440, 111.0437	3,4-Dicaffeoylquinic acid *	[43]
5	7.69	${\left[M+H\right]}^{+}$	0.368	C_25_H_32_ClN_5_O_7_	550.2063	550.2061	465.1531, 447.1421, 318.0839, 300.0741	Astin E	[43]
6	8.16	${\left[M+H\right]}^{+}$	0.544	C_25_H_33_N_5_O_8_	532.2408	532.2399	235.1075, 179.0841, 131.0491, 106.0655	Iso-asterinin A	[44]
7	8.53	${\left[M+H\right]}^{+}$	0.732	C_25_H_33_N_5_O_8_	532.2408	532.2398	235.1075, 131.0492, 106.0656	Asterinin A	[44]
8	9.09	${\left[M+H\right]}^{+}$	0.649	C_25_H_33_Cl_2_N_5_O_7_	586.1830	586.1826	558.2021, 251.0350, 131.0492, 106.0656	Astin A	[43]
9	9.53	${\left[M-H\right]}^{-}$	0.135	C_15_H_10_O_7_	301.0354	301.0353	301.0355, 178.9977, 151.0025	Quercetin *	[43]
10	9.67	${\left[M+H\right]}^{+}$	0.339	C_25_H_33_N_5_O_7_	516.2453	516.2451	338.1709, 235.1075, 193.0969, 179.0815	Astin J	[43]
11	9.97	${\left[M+H\right]}^{+}$	0.871	C_25_H_33_Cl_2_N_5_O_7_	570.1881	570.1876	485.1148, 163.0389, 131.0491, 106.0654	Astin C	[43]
12	10.61	${\left[M-H\right]}^{-}$	0.051	C_67_H_108_O_34_	1455.6637	1455.6639	1323.6244, 781.4380, 673.2198	Astersaponin A	[43]
13	11.03	${\left[M-H\right]}^{-}$	0.450	C_15_H_10_O_6_	285.0405	285.0406	257.0446, 151.0024	Kaempferol *	[43]
14	12.31	${\left[M-H\right]}^{-}$	3.491	C_18_H_34_O_5_	329.2326	329.2334	229.1442, 211.1334	Pinellic acid	[45]
15	21.57	${\left[M-H\right]}^{-}$	0.033	C_25_H_24_O_12_	279.2329	279.2330	279.2330	Linoleic acid *	[46]
16	22.41	${\left[M-H\right]}^{-}$	0.095	C_10_H_8_O_4_	281.2486	281.2487	281.2487	Oleic acid *	[46]

* Denotes comparison with authentic standards.

These findings suggest that the presence of these active phytochemicals in EERA may underlie its antiosteoclastogenic properties and beneficial effects against osteoporosis and metabolic alterations in OVX mice, potentially through synergistic mechanisms.

## 3. Discussion

The RANKL/RANK axis is pivotal in osteoclast differentiation and bone resorption during both physiological and pathological bone remodeling [3]. In the present study, we provided evidence that EERA possesses antiosteoclastogenic attributes by suppressing both RANKL expression and RANK-mediated osteoclastogenesis of precursor cells. Our in vivo evaluations further revealed the therapeutic potential of EERA in an OVX mouse model.

Previous studies have demonstrated that osteotropic factors such as VitD3, parathyroid hormone, and proinflammatory cytokines stimulate osteoclast differentiation in vitro by stimulating RANKL expression in osteoclast-supporting cells such as osteoblasts [3]. Although RANKL is expressed in a variety of cell types, including mesenchymal lineage cells like osteoblasts and immune cells [2], recent studies highlight that osteocytes predominantly serve as the main source of RANKL for in vivo osteoclastogenesis, providing it in a membrane-bound form [5,47]. Our results show that EERA dampens VitD3-driven RANKL mRNA expression in MLO-Y4 cells, an established osteocyte-like cell line [48], which subsequently reduces VitD3-induced osteoclast formation in BMM and MLO-Y4 cocultures.

Further, our findings underscore that EERA not only attenuates RANKL induction but also disrupts RANKL-mediated osteoclastogenesis in BMMs by modulating downstream RANK signaling pathways. We observed that EERA inhibits the RANKL-mediated upregulation of c-Fos and NFATc1 proteins during osteoclastogenesis. Given c-Fos’s pivotal role as an upstream regulator for NFATc1 induction [9], the antiosteoclastic properties of EERA likely stem from its ability to suppress c-Fos protein expression. Interestingly, while there was a notable reduction in c-Fos protein levels with EERA treatment, the RANKL-induced c-Fos mRNA levels remained unaffected. This pattern mirrors findings from a previous study, where interferon-β was shown to downregulate RANKL-mediated c-Fos protein expression through post-transcriptional pathways [49]. Delving deeper into EERA’s influence on the early RANKL-induced activation of MAPKs and the NF-κB pathway, our data revealed a significant inhibition of JNK and p38 MAPK activation by EERA, while ERK activation was enhanced. Additionally, EERA hindered the degradation of IκBα. Despite the well-documented roles of JNK, p38 MAPKs, and the NF-κB pathway in RANKL-driven osteoclastogenesis [28,29,30], ERK signaling presents a more complex picture, imparting both stimulatory and inhibitory impacts on osteoclastogenesis, potentially in a cell-type- or stage-specific manner [27]. The precise mechanism through which EERA modulates RANK signaling and its consequent effect on RANKL-induced c-Fos protein expression warrants further investigation.

Postmenopausal osteoporosis, a consequence of estrogen deficiency following menopause, presents a substantial global health challenge. This condition frequently coincides with other physiological changes, such as elevated body weight, visceral fat accumulation, and fatty liver development [50]. Although various therapeutic strategies exist, they often lack holistic efficacy and are accompanied by side effects [50,51]. Our research delved into the bone-protective potential of EERA using the OVX mouse model, which closely mirrors the hallmarks of postmenopausal osteoporosis, including decreased bone density and structural alterations [52]. Furthermore, the OVX model simulates the weight gain typically seen in postmenopausal women, emphasizing the increase in fat mass [31]. EERA treatment significantly ameliorated ovariectomy-induced trabecular bone loss and its microstructural changes. Moreover, its impact extended to mitigating weight gain and fat accumulation. Notably, EERA reduced ovariectomy-induced enlargement of the thymus and spleen. Enhanced sizes of these immune organs in OVX mice have been linked to T cell proliferation and TNF-α production [32], which augment osteoclast differentiation and consequent bone loss [53]. Further investigations are warranted to determine whether EERA’s antiosteoporotic effects are related to the reduction in the size of these organs. In our study, both the 30 and 100 mg/kg/day doses of EERA showed similar bone-protective effects, which contrasts with the dose-responsive patterns observed for other metabolic parameters such as weight gain and perigonadal fat weight. Our phytochemical analysis of EERA identified several phytochemical compounds known to influence both bone and fat metabolism. However, further research is needed to identify the specific fractions or constituents within EERA responsible for its beneficial effects, which could provide insights into its dose-independent mechanism of protecting bone loss, distinct from its dose-dependent effects on other metabolic parameters.

Separately, while Radix Asteris possesses numerous beneficial attributes, reports have indicated potential liver toxicity. In an acute toxicity assessment, a 75% ethanol extract of Radix Asteris, akin to our EERA formulation, exhibited an LD50 of 15.74 g/kg in mice, predominantly causing hepatic toxicity without affecting other organs [54]. A subchronic toxicity evaluation employing a 350 mg/kg/day dose over 91 days in rats revealed hepatic lesions and elevated serum ALT and AST levels in the petroleum ether fraction but not the 75% ethanol extract itself, suggesting the toxic elements are concentrated in the petroleum ether fraction [54]. Our EERA preparation contains unique Radix Asteris components, including cyclic astins and linear asterinins [13,14]. Some of these compounds, notably astin B and C, have been considered hepatotoxic agents from Radix Asteris [55,56]. Nevertheless, our results indicated that EERA, at a dose of 100 mg/kg/day, significantly reduced elevated serum ALT levels in OVX mice, suggesting that the tested EERA dosages might not elicit hepatic toxicity.

## 4. Materials and Methods

### 4.1. Materials

Fetal bovine serum (FBS) and α-minimal essential medium (α-MEM) were sourced from Thermo Fisher Scientific (Waltham, MA, USA). VitD3 was purchased from Sigma-Aldrich (St. Louis, MO, USA). Recombinant M-CSF and RANKL were obtained as detailed in a previous report [57], and the MLO-Y4 cell line was procured from Kerafast (Boston, MA, USA).

### 4.2. EERA Preparation

For the EERA powder preparation, Radix Asteris (0.5 g) was extracted with a 7-fold volume of distilled water at 100 °C for 3 h under reflux conditions. Post extraction, the mixture was filtered, and the filtrate was lyophilized to yield the EERA powder. Before experimental use, this powder was dissolved in distilled water and filtered through a 0.2 μm membrane to remove any particulates.

### 4.3. BMM Culture and Cytotoxicity Assay

BMMs were prepared and cultured in α-MEM complete medium, which consisted of 10% heat-inactivated FBS and 1% penicillin/streptomycin, supplemented with M-CSF (60 ng/mL) as described previously [21]. For assessing EERA’s potential cytotoxic effects on BMMs, cells were treated with varying concentrations of EERA for a duration of 24 h. Following this incubation, cell viability was determined using the Cell Counting Kit-8 assay as per the manufacturer’s guidelines (Dojindo Molecular Technologies Inc., Rockville, MD, USA).

### 4.4. Osteoclast Differentiation Assay

For assessing the potential effects of EERA on osteoclast differentiation, both coculture and single BMM culture systems were employed. In the coculture setup, BMMs (4 × 10^4^ cells/well) and MLO-Y4 cells (1 × 10^3^ cells/well) were seeded in 96-well plates using α-MEM complete medium. Differentiation was induced by treating the cocultures either with VitD3 (10 nM) for a duration of 5 days or a combination of VitD3 (10 nM) and RANKL (50 ng/mL) for 4 days. These treatments were administered in both the absence and presence of varying concentrations of EERA, ranging from 33.3 to 200 μg/mL. For assessing osteoclast differentiation in isolated BMMs, the cells (1 × 10^4^ cells/well) were seeded in 96-well plates. Differentiation was induced using M-CSF (60 ng/mL) and RANKL (50 ng/mL) for 4 days, with or without the addition of EERA at concentrations ranging from 33.3 to 200 μg/mL in α-MEM complete medium. To quantify osteoclast formation, post-treatment cells were fixed and permeabilized using 0.1% Triton X-100. TRAP staining was then conducted as described previously [21].

### 4.5. Real-Time PCR Analysis

For the analysis of gene expression, total RNA was isolated from treated MLO-Y4 cells or BMMs using the RNeasy Mini kit (Qiagen, Hilden, Germany). Subsequently, complementary DNA (cDNA) was synthesized utilizing 1 μg of the extracted RNA with the aid of a cDNA reverse transcription kit (Thermo Fisher Scientific). The amplification of specific target mRNAs was carried out utilizing the TaqMan Universal Master Mix II with TaqMan probes on an ABI 7500 Real-Time PCR system (Applied Biosystems, Foster City, CA, USA). The following TaqMan probes were employed for target gene amplification: Tnfsf11, Mm00441908_m1; Tnfrsf11b, Mm00435454_m1; Csf1, Mm00432686_m1; c-Fos, Mm00487425_m1; NFATc1, Mm00479445_m1; Atp6v0d2, Mm00656638_m1; Tm7sf4, Mm01168058_m1; Ctsk, Mm00484036_m1; Irf8, Mm00492567_m1; Prdm1, Mm00476128_m1; 18S rRNA, Hs99999901_s1. For normalization, 18S rRNA was used as an internal reference gene. Gene expression was quantified using the ∆∆Ct method.

### 4.6. Western Blot Analysis

For protein extraction, BMMs were lysed using a lysis buffer (PRO-PREPTM; iNtRON biotechnology, Sungnam, Republic of Korea). Equal amounts of protein samples were denatured by heating at 100 °C for 5 min and then subjected to sodium dodecyl sulfate-polyacrylamide gel electrophoresis for separation. Subsequently, the separated proteins were transferred from the polyacrylamide gels onto polyvinylidene fluoride membranes (Bio-Rad Laboratories, Hercules, CA, USA). The transferred membranes were blocked with 5% skim milk in TBS-T for 2 h, followed by an overnight incubation at 4 °C with the primary antibodies. Primary antibodies targeting phosphorylated forms like p-JNK (T183/Y185), p-p38 (T180/Y182), p-ERK (T202/Y204), and p-IκBα (S32), as well as their nonphosphorylated counterparts, were obtained from Cell Signaling Technology (Danvers, MA, USA). Other antibodies, including NFATc1 and c-Fos, were sourced from Santa Cruz Biotechnology (Dallas, TX, USA). Following three washes with TBS-T, the target protein bands were detected using secondary antibodies conjugated to horseradish peroxidase and a chemiluminescent substrate (Thermo Fisher Scientific) on a ChemiDoc Touch imaging system (Bio-Rad Laboratories). All Western blots were performed in three independent experiments. The protein bands were quantified by densitometry using Image Lab software version 5.2.1 (Bio-Rad Laboratories).

### 4.7. Analysis of RANKL Protein Expression

To analyze RANKL protein expression, MLO-Y4 cells were lysed using a lysis buffer (PRO-PREPTM). Following lysis, the protein levels of RANKL in the cell lysates were quantified according to the manufacturer’s instructions (R&D Systems, Minneapolis, MN, USA).

### 4.8. Animal Experiments

Female C57BL/6J mice, aged six weeks, were maintained under standardized laboratory conditions with a standard chow diet. Following a one-week acclimatization, the mice underwent either sham surgery or bilateral ovariectomy. Subsequently, the mice were stratified into four groups, each comprising eight mice: sham, OVX, OVX treated with EERA at a dose of 30 mg/kg, and OVX treated with EERA at a dose of 100 mg/kg. Starting one week after the surgery, EERA was orally administered to the OVX mice once daily for a duration of 6 weeks, along with the provision of a purified normal fat rodent diet (D12450B, Research Diets, New Brunswick, NJ, USA). After completing the treatment period, following a fasting duration of 7 h, mice were euthanized for sample collection. Femoral bone, spleen, thymus, uterus, perigonadal fat, and serum were harvested for subsequent analyses. Serum ALT and AST levels were quantified using an automatic biochemical analyzer (Hitachi 7180; Hitachi, Tokyo, Japan).

### 4.9. μ-CT Analysis

The distal femur from the right leg of each mouse, as specified in the treatment groups, was scanned using a SkyScan 1276 μ-CT system (Bruker, Kontich, Belgium). The CTvol software (version 3.3.0r1383, Bruker) was employed to generate three-dimensional images of the distal femur metaphysis. Subsequently, the scanned images were reconstructed using Nrecon software (version 1.7.42). To assess bone quality, parameters including BMD and various trabecular bone morphometric parameters at the metaphyseal trabecular bone were quantitatively analyzed using CTAn software (version 1.20.3.0).

### 4.10. UHPLC-MS/MS Analysis

To characterize the components of EERA, UHPLC-MS/MS analysis was conducted, adopting methodologies previously described [57]. Reference standards of the phytochemicals, namely chlorogenic acid, 3,4-dicaffeoylquinic acid, quercetin, kaempferol, linoleic acid, and oleic acid, were procured from Targetmol (Wellesley Hills, MA, USA). For the UHPLC-MS/MS procedure, a Dionex UltiMate 3000 system interfaced with a Thermo Q-Exactive mass spectrometer was employed. The chemical constituents were separated via liquid chromatography on an Acquity BEH C18 column (100 × 2.1 mm, 1.7 μm). A gradient elution for the mobile phase was established, consisting of 0.1% formic acid in water (A) and acetonitrile (B): 0–1 min, 3% B; 1–2 min, 3–15% B; 2–13 min, 15–50% B; 13–20 min, 50–100% B; 20–23 min, 100% B; and 23.5–27.5 min, 3% B. The Q-Exactive mass spectrometer was operated in both negative and positive ion switching modes. Data acquisition and analysis were conducted using Xcalibur (version 4.1) and TraceFinder software (version 4.0, Thermo Fisher Scientific).

### 4.11. Statistical Analysis

Data analysis was conducted using GraphPad Prism (version 9.5.1). Data from in vitro experiments are presented as the mean ± standard deviation, whereas in vivo study results and Western blot analyses are expressed as the mean ± standard error of the mean. Group differences were assessed using one-way or two-way ANOVA, followed by either Dunnett’s or Sidak’s post hoc tests as appropriate.

## 5. Conclusions

In summary, our research provides evidence of EERA’s inhibitory effects on osteoclastogenesis and elucidates the underlying mechanisms, emphasizing its potential as a bone-protective agent in OVX mice. Beyond its osteoprotective effects, EERA’s influence on other physiological alterations induced by estrogen deficiency, such as weight gain, fat accumulation, and liver function, underscores its potential as a promising candidate for addressing postmenopausal osteoporosis and its accompanying physiological changes. Further in-depth studies on EERA’s impact on bone cells and its safety profile, especially concerning hepatotoxicity, are crucial before advocating its use for postmenopausal women.

## Figures and Tables

**Figure 1 ijms-24-16526-f001:**
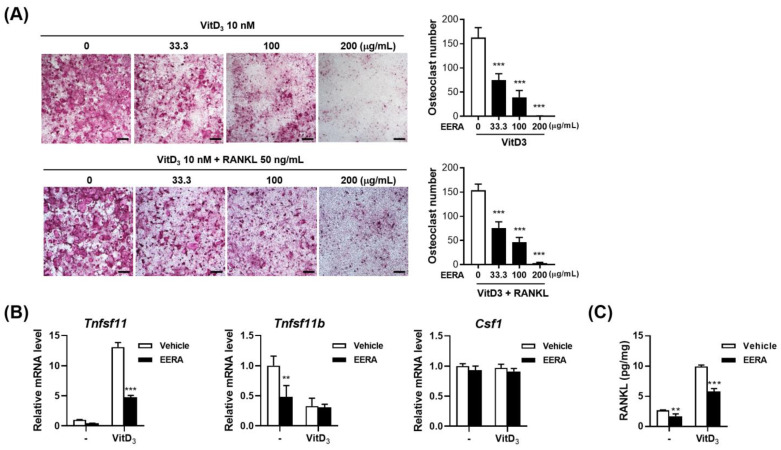
Influence of EERA on osteoclastogenesis in the MLO-Y4/BMMs coculture system. (**A**) Coculture of MLO-Y4 and BMMs was treated with vehicle or graded concentrations of EERA in the presence of 10 nM 1α,25-dihydroxyvitamin D3 (VitD3) for 5 days (**upper panel**), or combined VitD3 + RANKL (50 ng/mL) for 4 days (**lower panel**). Displayed are representative microscopy images from TRAP staining (scale bar, 100 µm) alongside the osteoclast quantification per well. (**B**,**C**) MLO-Y4 cells, exposed to EERA (200 μg/mL) and/or VitD3 (10 nM) for a day, were analyzed for mRNA levels of *Tnfsf11*, *Tnfsf11b*, and *Csf1* using real-time PCR (**B**) and RANKL protein levels in cell lysates using an ELISA kit (R&D Systems, Minneapolis, MN, USA). ** *p* < 0.01; *** *p* < 0.001 versus the vehicle control.

**Figure 2 ijms-24-16526-f002:**
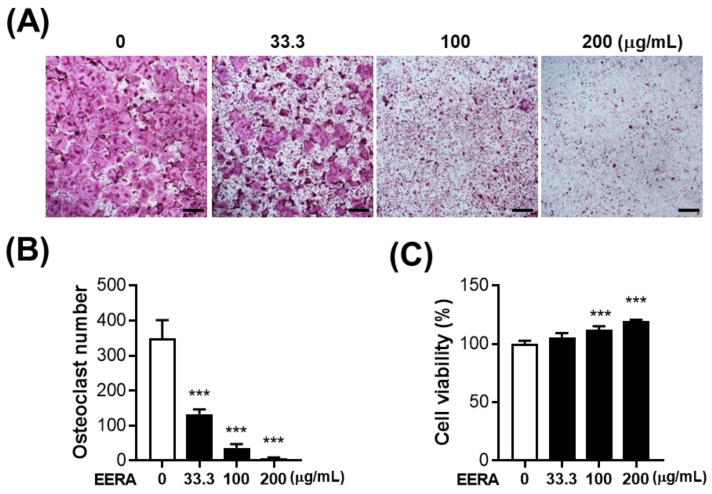
Influence of EERA on RANKL-induced osteoclastogenesis. (**A**) BMMs were cultured within M-CSF and RANKL across a gradient of EERA concentrations (0–200 μg/mL). Displayed are representative images of TRAP staining (scale bar, 100 µm). (**B**) Quantitative analysis of osteoclasts per well. (**C**) Cell viability assessment in BMMs post a 1-day treatment with varying EERA concentrations (0–200 μg/mL). *** *p* < 0.001 compared with the vehicle control.

**Figure 3 ijms-24-16526-f003:**
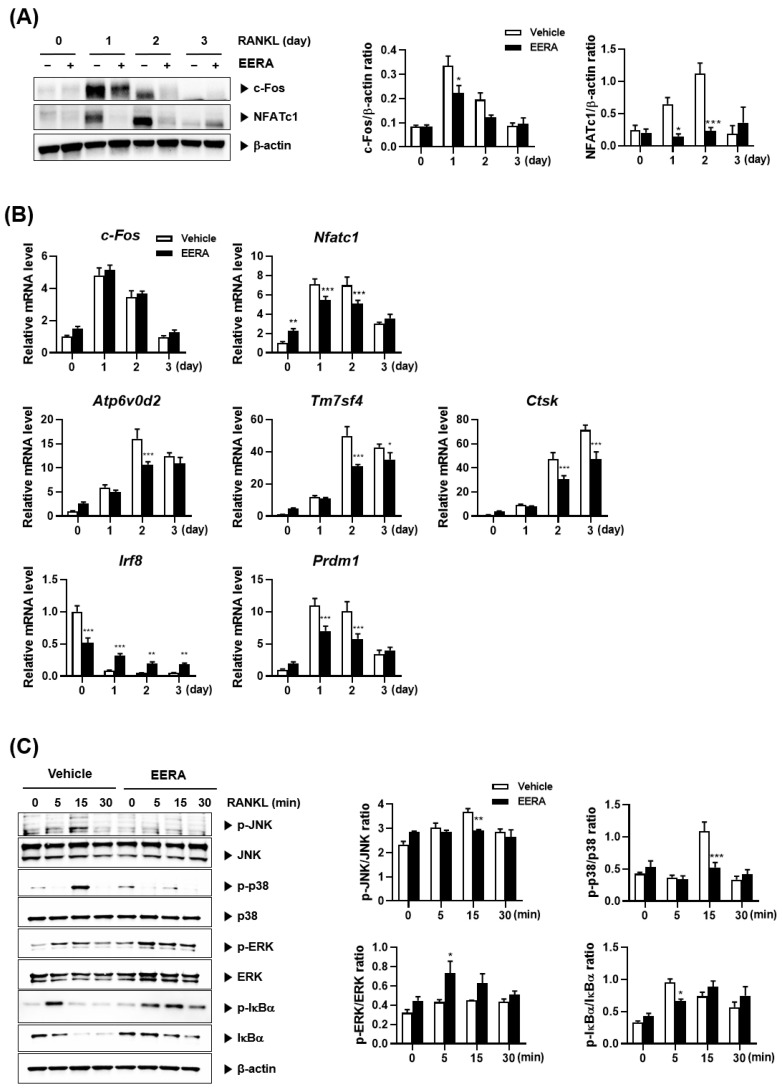
Modulation of RANKL-induced signaling pathways by EERA. (**A**,**B**) BMMs were treated with or without EERA (200 μg/mL) and RANKL (50 ng/mL) for the specified durations. Protein levels (**A**) and mRNA expression (**B**) of the target molecules were assessed using Western blotting and real-time PCR, respectively. (**C**) Western blot analysis of phosphorylated and nonphosphorylated forms of JNK, p38, ERK, and IκBα. Quantified Western blot data are shown as the means ± standard error of the mean of three separate experiments. * *p* < 0.05; ** *p* < 0.01; *** *p* < 0.001 versus the vehicle control.

**Figure 4 ijms-24-16526-f004:**
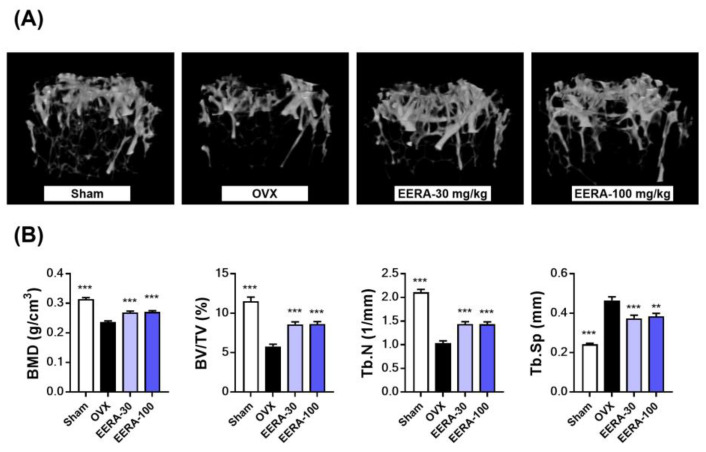
Impact of EERA on bone loss in OVX mice. Postovariectomy, mice received EERA orally at either 30 mg/kg/day (EERA-30) or 100 mg/kg/day (EERA-100) starting 1 week after surgery and continuing for 6 weeks. (**A**) Three-dimensional reconstructed μ-CT images of trabecular bone in the distal femur metaphysis. (**B**) Quantitative assessment of BMD and femoral metaphyseal trabecular bone morphometrics. ** *p* < 0.01; *** *p*  <  0.001 versus the OVX control.

**Figure 5 ijms-24-16526-f005:**
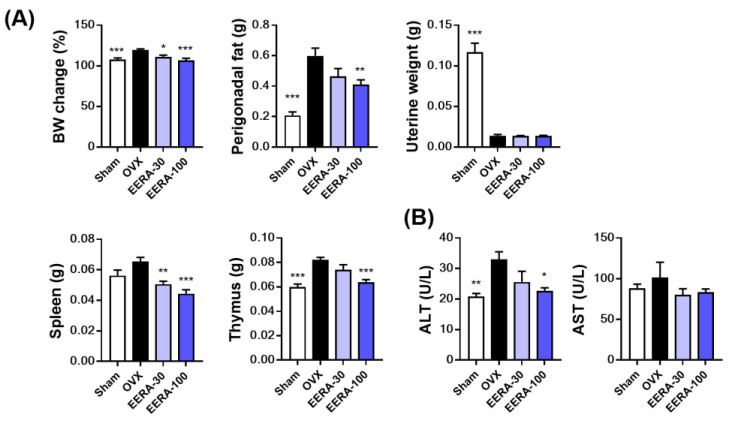
EERA’s influence on metabolic alterations in OVX mice. Postovariectomy, mice received EERA orally at either 30 mg/kg/day (EERA-30) or 100 mg/kg/day (EERA-100) starting 1 week after surgery and continuing for 6 weeks. (**A**) Quantitative evaluation of body weight, perigonadal fat weight, and weights of uterine, spleen, and thymus. (**B**) Analysis of serum ALT and AST concentrations. * *p* < 0.05; ** *p* < 0.01; *** *p* < 0.001 versus the OVX group.

**Figure 6 ijms-24-16526-f006:**
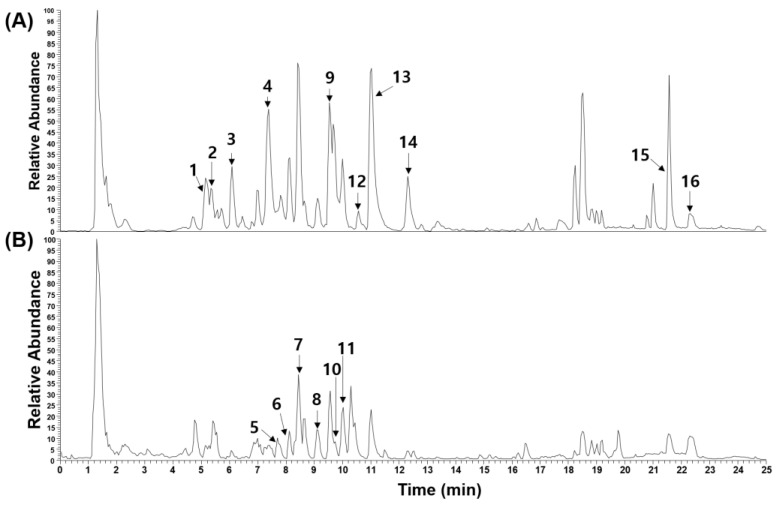
Phytochemical analysis of EERA. Base peak chromatograms obtained under negative (**A**) and positive (**B**) ionization modes. Identified compounds are denoted by peak numbers: 1, chlorogenic acid; 2, feruloylquinic acid 1; 3, feruloylquinic acid 2; 4, 3,4-dicaffeoylquinic acid; 5, astin E; 6, iso-asterinin A; 7, asterinin A; 8, astin A; 9, quercetin; 10, astin J; 11, astin C; 12, astersaponin A; 13, kaempferol; 14, pinellic acid; 15, linoleic acid; 16, oleic acid.

## Data Availability

Data are contained within the article.

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
