# Peer review of "Ethanol Extract of Radix Asteris Suppresses Osteoclast Differentiation and Alleviates Osteoporosis"

_ijms, 2023, doi:10.3390/ijms242216526_

Round 1
Reviewer 1 Report
Comments and Suggestions for Authors
Review note
This paper studies the protective effects of ethanol extract of Radix Asteris (EERA) against osteoporosis by suppressing osteoclast differentiation. I appreciate the authors’ works, which will help further (pre)clinical practice. However, to grow into a publication, I think there are some issues the authors need to address.
1. The authors used ethanol extract of Radix Asteris (EERA) throughout the contents but used Radix Asteris Root Extract in the title. Any particular reason for not using EERA for the consistency?
2. The authors first used MLO-Y4/BMMs co-culture System to test the effect of EERA in OCs in their first experiment. This experiment suggested EERA may also affect osteocytes in their RANKL and OPG expression. These should be confirmed by ELISA. Also, Ooteoblasts mainly regulates osteoclasts through RANKL/OPG system. Does EERA also affect OB secretion? This could be easily addressed by using OB-OC co-culture system.
3. How to understand that EERA significantly inhibit the osteoclasts to almost 1/4 to 1/20 of the control level, meanwhile, increasing their cell viability in figure 2?
4. WB should be quantified in figure 3.
5. I suggest authors show trabecular bones only (either u-CT scan image or 3D reconstruction) rather than a complete 3D reconstruction image in figure 4A.
6. Minors: the scale bars in Fig1A are only in the same place of each image.
In summary, I feel the study is solid. However, improvement is needed. I hope the author(s) could find some of the above discussions helpful for improving the paper.
Comments on the Quality of English LanguageMinor editing of English language required
Author Response
Reviewer 1
This paper studies the protective effects of ethanol extract of Radix Asteris (EERA) against osteoporosis by suppressing osteoclast differentiation. I appreciate the authors’ works, which will help further (pre)clinical practice. However, to grow into a publication, I think there are some issues the authors need to address.
- The authors used ethanol extract of Radix Asteris (EERA) throughout the contents but used Radix Asteris Root Extract in the title. Any particular reason for not using EERA for the consistency?
Response: Thank you for your insightful observation. Following your suggestion, we have revised the title to now read "Ethanol Extract of Radix Asteris (EERA)," aligning it with the term used throughout the text.
- The authors first used MLO-Y4/BMMs co-culture System to test the effect of EERA in OCs in their first experiment. This experiment suggested EERA may also affect osteocytes in their RANKL and OPG expression. These should be confirmed by ELISA. Also, Ooteoblasts mainly regulates osteoclasts through RANKL/OPG system. Does EERA also affect OB secretion? This could be easily addressed by using OB-OC co-culture system.
Response: Thank you for your valuable feedback. In response to your suggestion, we have extended our investigation to include the effects of EERA on RANKL protein expression, utilizing ELISA for analysis. The results, which we have now included in Figure 1C of the revised manuscript, revealed that EERA markedly diminished both the basal and VitD3-stimulated RANKL protein expression in MLO-Y4 cell lysates, consistent the observed decrease in RANKL mRNA levels. However, we encountered challenges in detecting RANKL and OPG secretion in the supernatants of MLO-Y4 cultures using ELISAs, as these levels were below the detection limit of the assays.
As discussed in our manuscript (lines 252-255), osteocytes are thought to be primary sources of RANKL, which guided our initial experimental design. Nonetheless, we recognize the relevance of exploring EERA's impact on osteoblasts as well. Unfortunately, due to current restrictions at our institute, we are unable to obtain primary osteoblasts for further experimentation.
- How to understand that EERA significantly inhibit the osteoclasts to almost 1/4 to 1/20 of the control level, meanwhile, increasing their cell viability in figure 2?
Response: In Figure 2A and B, we demonstrate that EERA, at concentrations between 33.3 and 200 μg/mL, significantly inhibits RANKL-induced differentiation of osteoclast precursor cells, BMMs. Figure 2C shows that, within the same concentration range, EERA does not exert cytotoxic effects on BMMs. To more clearly describe these findings, we have revised the manuscript to include an expanded explanation in lines 123-125 as follows: “indicating that the anti-osteoclastogenic effect of EERA is not due to cytotoxicity on osteoclast precursor cells.”
- WB should be quantified in figure 3.
Response: We appreciate your suggestion regarding the quantification of WB results in Figure 3. Following your recommendation, we have now quantified the WB data and updated Figure 3 accordingly.
- I suggest authors show trabecular bones only (either u-CT scan image or 3D reconstruction) rather than a complete 3D reconstruction image in figure 4A.
Response: We are grateful for your insightful recommendation to emphasize the trabecular bones in Figure 4A. Following your advice, we have updated the figure to display solely the 3D reconstructed images of trabecular bone.
- Minors: the scale bars in Fig1A are only in the same place of each image.
Response: Thank you for pointing out the inconsistency in the placement of scale bars in Figure 1A. We have now amended this issue by positioning the scale bars uniformly across all images in Figure 1A.

Reviewer 2 Report
Comments and Suggestions for Authors
The manuscript presents compelling results that an extract from RADIX Asteris Root, designated EERA, suppresses osteoclast differentiation. Mechanistically, findings support involvement of RANKL signaling. In vitro as well as in vivo data is provided. There is potential for developing EERA for treatment of post-menopausal osteoporosis. Biological and regulatory activity are examined with data secured from EERA influence on gene expression that is functionally linked to control of bone remodeling and skeletal homeostasis. However, while phytochemical analysis of EERA is presented, that is consistent with mediators of bone biology and pathology, the current studies are limited by the absence of assessment for EERA fractions and lack of clarity on concentrations and dose response.
Author Response
Reviewer 2
The manuscript presents compelling results that an extract from RADIX Asteris Root, designated EERA, suppresses osteoclast differentiation. Mechanistically, findings support involvement of RANKL signaling. In vitro as well as in vivo data is provided. There is potential for developing EERA for treatment of post-menopausal osteoporosis. Biological and regulatory activity are examined with data secured from EERA influence on gene expression that is functionally linked to control of bone remodeling and skeletal homeostasis. However, while phytochemical analysis of EERA is presented, that is consistent with mediators of bone biology and pathology, the current studies are limited by the absence of assessment for EERA fractions and lack of clarity on concentrations and dose response.
Response: We appreciate the insightful feedback and have accordingly expended the Discussion section of our manuscript to more directly address the limitations and the necessity of assessing EERA fractions and clarifying dose responses as follows (lines 293-301): “In our study, both the 30 and 100 mg/kg/day doses of EERA showed similar bone-protective effects, which contrasts with the dose-responsive patterns observed for other metabolic parameters such as weight gain and perigonadal fat weight. Our phytochemical analysis of EERA identified several phytochemical compounds known to influence both bone and fat metabolism. However, further research is needed to identify the specific fractions or constituents within EERA responsible for its beneficial effects, which could provide insights into its dose-independent mechanism of protecting bone loss, distinct from its dose-dependent effects on other metabolic parameters.”

Round 2
Reviewer 2 Report
Comments and Suggestions for Authors
Manuscript is now acceptable for publication